# Validity of Current Smartwatches for Triathlon Training: How Accurate Are Heart Rate, Distance, and Swimming Readings?

**DOI:** 10.3390/s24144675

**Published:** 2024-07-18

**Authors:** Tobias Jacko, Julia Bartsch, Carlo von Diecken, Olaf Ueberschär

**Affiliations:** 1Department for Health Sciences, Medicine and Research, Faculty of Health and Medicine, Danube University Krems, 3500 Krems an der Donau, Austria; tobias.jacko@edu.donau-uni.ac.at; 2Department of Engineering and Industrial Design, Magdeburg-Stendal University of Applied Sciences, 39114 Magdeburg, Germany; julia.bartsch@h2.de (J.B.);; 3Department of Biomechanics and Sports Technology, Institute for Applied Training Science, 04109 Leipzig, Germany

**Keywords:** wearables, sports watch, reliability, validity, GNSS, swimming, running, cycling

## Abstract

Smartwatches are one of the most relevant fitness trends of the past two decades, and they collect increasing amounts of health and movement data. The accuracy of these data may be questionable and requires further investigation. Therefore, the aim of the present study is to validate smartwatches for use in triathlon training. Ten different smartwatches were tested for accuracy in measuring heart rates, distances (via global navigation satellite systems, GNSSs), swim stroke rates and the number of swim laps in a 50 m Olympic-size pool. The optical heart rate measurement function of each smartwatch was compared to that of a chest strap. Thirty participants (15 females, 15 males) ran five 3 min intervals on a motorised treadmill to evaluate the accuracy of the heart rate measurements. Moreover, for each smartwatch, running and cycling distance tracking was tested over six runs of 4000 m on a 400 m tartan stadium track, six hilly outdoor runs over 3.4 km, and four repetitions of a 36.8 km road bike course, respectively. Three swimming protocols ranging from 200 m to 400 m were performed in triplicate in a 50 m Olympic-size pool, evaluating the tracked distance and the detected number of strokes. The mean absolute percentage errors (MAPEs) for the average heart rate measurements varied between 3.1% and 8.3%, with the coefficient of determination ranging from 0.22 to 0.79. MAPE results ranged from 0.8% to 12.1% for the 4000 m run on the 400 m track, from 0.2% to 7.5% for the 3.4 km outdoor run, and from 0.0% to 4.2% for the 36.8 km bike ride. For the swimming tests, in contrast, the deviations from the true distance varied greatly, starting at a 0.0% MAPE for the 400 m freestyle and reaching 91.7% for the 200 m medley with style changes every 25 m. In summary, for some of the smartwatches, the measurement results deviated substantially from the true values. Measurements taken while road cycling over longer distances with only a few curves were in relative terms more accurate than those taken during outdoor runs and even more accurate than those taken on the 400 m track. In the swimming exercises, the accuracy of the measured distances was severely deteriorated by the medley changes among the majority of the smartwatches. Altogether, the results of this study should help in assessing the accuracy and thus the suitability of smartwatches for general triathlon training.

## 1. Introduction

Since 2016, the American College of Sports Medicine has listed wearable technology as one of the top three fitness trends worldwide, demonstrating the enormous impact wearables are having on the sports community [1,2,3,4,5,6,7,8]. Using photoplethysmography, a key sensing technology, it is possible to track physiological parameters such as heart rate, heart rate variability, heart rhythm, pulse wave, respiratory rate, oxygen saturation, and sleep quality [9]. The technical possibilities continue to increase, as wearables are becoming progressively smaller, and better sensors, such as accelerometers, gyroscopes, or global navigation satellite system (GNSS) receiver chips, are being built into the watches. This increases the comfort for an athlete who wishes to track their training automatically with virtually no additional technical effort. On the other hand, external sensors, such as temperature sensors or additional acceleration sensors, are also being included in wrist, trunk or shoe pods to record additional data. In 2022, 492 million wearables were sold worldwide, with a predicted upward trend [10]. Today, wearable technology offers the advantage of balancing training load and monitoring health status at the same time. Athletes, coaches, and sports scientists have identified its potential to prevent injuries, improve performance, and prolong an athlete’s career. However, the validity, accuracy, and reliability of this evolving technology may be, in parts, controversial and uncertain [11]. In a survey of triathlon coaches, it was found that some coaches were concerned about athletes’ overreliance on technology and that the data it produces can impact their perceptions on performance [12]. The enormous potential and constant development of smartwatch technology regularly necessitates the review of current and new models. Various models have already been assessed, however, not with a triathlon-specific focus in mind [13,14,15,16,17,18]. Notwithstanding, the accuracy of the measured parameters is crucial to achieve the desired benefits.

Hence, the goal of the present study was to determine the measurement accuracy of ten different sports watches in regard to optical heart rate (i.e., pulse rate) measurements and GNSS-based distance tracking, as well as in the acquisition of swimming data. The following questions were addressed: How accurate is the optical heart rate measurement during a 3 min run on a treadmill? How accurate are the GNSS measurements during a 4000 m run on a 400 m tartan stadium track, a 3.4 km hilly outdoor run, and a 36.8 km profiled bike ride? Are there differences in the GNSS measurements among the various watches? Furthermore, do such differences in the measured distances depend on whether the watch is worn on the wrist facing the inside or outside of the running track/course curvature? Lastly, how accurate are the lap counts and arm stroke counts in a 50 m Olympic-size swimming pool? These questions focus on typical single-discipline triathlon training, which is usually divided into separate sessions of swimming, cycling, and/or running. An assessment for use in brick sessions or competitions was not carried out, as it would require a multisport function, which is not supported by all of the selected smartwatches.

## 2. Materials and Methods

### 2.1. Smartwatches

Ten different smartwatches currently available on the market (as of May 2024) from nine manufacturers were included in this study (Table 1). The smartwatches were selected to cover a wide range of manufacturers with adequate market shares, ranging from entry-level products at a recommended retail price of 120 €, such as the Mi Watch (XIA) (Xiaomi, Inc., Beijing, China), to an advanced triathlon-specific model, the Garmin Forerunner^®^ 955 Solar (GAF) (Garmin AG, Schaffhausen, Switzerland), at 650 €. In an overview of global sales figures for wearables from 2014 to 2022, in total, eight leading manufacturers were listed, seven of which are included in this study: Apple, Samsung, Xiaomi, Huawei, Fitbit, Garmin, and Fossil (order according to rank) [19]. Looking at the most popular smartwatch brands in Germany in 2023, the top seven are represented in this study, supplemented by Fossil in rank 10. Moreover, Polar was included in this study, owing to the popularity of their smartwatches in Germany [19] and because Polar watches were analysed in previous studies [15,16,18].

As regards movement tracking, a multi-GNSS analysis is more accurate than a single GNSS [20]. For this reason, the GNSS function of the watches was set to utilise both the American GPS and the Russian GLONASS systems in parallel to achieve better comparability where applicable. If this was not available, the watch’s default settings (commonly GPS only) were used. The possible GNSS options are shown in Table 1, based on the information from the respective manufacturers. Further hardware specifications can be found in Table A5. Notably, Fossil does not specify the supported GNSSs in the Fossil Gen 6 Smartwatch (FOS) (Fossil, Inc., Richardson TX, USA). However, as its Snapdragon Wear 4100/4100+ (Qualcomm Technologies, Inc., San Diego CA, USA) processor supports GPS, GLONASS, Galileo, and BeiDou (BDS), that smartwatch is expected to perform accordingly [21]. To compensate for any variability in production, GNSS data were simultaneously collected using two separate watches of the same model, worn unilaterally by the same individual when running or mounted to the handlebar when cycling. During running and swimming, it was double-checked for each smartwatch that it was firmly attached and tightly worn on the athlete’s wrist, ensuring direct skin contact and minimising any relative motion between watch and wrist (Figure 1).
sensors-24-04675-t001_Table 1Table 1Overview of the ten investigated smartwatches, their manufacturers, and available GNSS options.SmartwatchManufacturerAbbreviationGNSS Options
GTS3Amazfit (Zepp North America, Inc. Irvine, CA, USA)AMAGPS, GLONASS, Galileo, BDS, QZSS[22,23]Watch SEApple, Inc. (Cupertion, CA, USA)APPGPS, GLONASS, Galileo, QZSS[24]Versa 4Fitbit, Inc. (San Francisco, CA, USA)FITGPS, GLONASS[25]Gen 6 SmartwatchFossil, Inc. (Richardson TX, USA)FOSGPS, GLONASS, Galileo, BDS[21,26,27]Forerunner^®^ 955 SolarGarmin AG (Schaffhausen, Switzerland)GAFGPS, GLONASS, Galileo, BDS, QZSS, IRNSS[28,29,30]Venu^®^ 2Garmin AG (Schaffhausen, Switzerland)GAVGPS, GLONASS, Galileo[31,32]Watch GT 3Huawei (Shenzhen, China)HUAGPS, GLONASS, Galileo, BDS, QZSS[33,34]Ignite 2Polar, Inc. (Kempele, Finnland)POLGPS, GLONASS, Galileo, QZSS[35,36]Galaxy Watch 4Samsung Electronics Co., Ltd. (Seoul, South Korea)SAMGPS, GLONASS, Galileo, BDS[37,38]Mi WatchXiaomi, Inc. (Beijing, China)XIAGPS, GLONASS, Galileo, BDS[39]GNSS: Global Navigation Satellite System; GPS: Global Positioning System (USA); GLONASS: Globalnaya Navigazionnaya Sputnikovaya Sistema (Russia); BDS: BeiDou Navigation Satellite System (China); QZSS: Quasi-Zenith Satellite System (Japan); IRNSS: Indian Regional Navigation Satellite System (India).

For all measurements and watches, the most suitable sport mode for the planned measurement was selected; for running, this was either outdoor running or track running. For watches that did not provide a specific track running mode, the regular outdoor running mode was selected. For road cycling, the outdoor cycling mode was used. If the watch did not offer this mode, the outdoor running mode was chosen instead, which was necessary with the FOS watch. Because of the higher average movement speed in cycling, it can be assumed that this differing choice of mode did not significantly influence the cycling distance measurements. For swimming, the indoor swimming mode was selected for any watch, and the lap length was set to 50 m. If a software update for a device or its corresponding application became available during the measurement period, it was installed to provide the most current device software status available.

### 2.2. Participants

A total of 30 individuals (15 males/15 females) with various training backgrounds, ranging from recreational athletes to trained long-distance triathletes (age: 29.4 ± 7.4 years; height: 175.6 ± 8.8 cm; body mass: 70.2 ± 8.2 kg), participated in the heart rate measurement part of the study (Table 2). For the running, cycling, and swimming parts, nine trained athletes of this cohort were recruited because the study design allowed for multiple measurements to be completed by the same individual without the loss of statistical power. The measurements were conducted in and around Leipzig, Germany, and Magdeburg, Germany, from April to June 2023.

### 2.3. Heart Rate Measurements

The arterial blood pressure in rest was measured manually and bilaterally with a sphygmomanometer (boso med 1, Bosch + Sohn GmbH & Co. KG, Jungingen, Germany) by a physician. The resting heart rate was determined by electrocardiography (ECG), which was also used to rule out any deviations in the cardiac currents that could have led to an incorrect measurement. Except for two extrasystoles for one subject, which did not lead to exclusion due to their statistical insignificance, no abnormalities were detected that could have contributed false measurements. In particular, heart rate and pulse rate could be treated as equivalents for all subjects. In addition, the resting ECG and blood pressure measurements were examined to detect any pathologies that would have contraindicated a stress test. If the ECG results were normal and the systolic blood pressure was ≤160 mmHg, the treadmill test was started.

For this purpose, the participants completed 5 × 3 min intervals at a self-selected running speed (min.: 8 km/h) on a motorised treadmill (h/p/cosmos saturn^®^, 250/100, h/p/cosmos sports and medical GmbH, 83365 Nußdorf–Traunstein, Germany or Star Trac 10 FreeRunner™, MERCOR Fitnesskonzepte GmbH Leipzig, Germany, respectively). An 1.5% incline was set to compensate for the absence of air resistance. While running, optical heart rate data (i.e., pulse rate data) were collected continuously by two different smartwatch models in parallel using their photoplethysmography sensors, with one watch being worn on each wrist. Therefore, the total number of intervals to be run for 10 smartwatches could be reduced from 10 to 5. As reference, true heart rate was measured electronically using a chest strap heart rate monitor (Garmin HRM-PRO, Garmin AG, Schaffhausen, Switzerland). The validity of the chest strap measurement has been confirmed in previous studies [40,41,42,43]. The measurements using the smartwatches and the chest strap started simultaneously and ended after 3 min of running at the target speed. All heart rate measurements were conducted with a data rate of at least 1 Hz.

### 2.4. Tracking of Running and Cycling Distances

To evaluate the accuracy of the distance tracking measurements, the smartwatches were tested on reference routes for running and cycling. To avoid environmental influences [44], the same location and the same date and time were used under a clear sky. Before the GNSS measurements were performed, a bike computer (Sigma BC 8.12, SIGMA-ELEKTRO GmbH, Neustadt, Germany) was calibrated with the tyre size. The rolling length of one tyre’s circumference was measured by marks on the ground while the rider was sitting on the bike (tyre size: 25 × 700c at 7.5 bar/109 PSI). This calibration was crosschecked and confirmed by measuring the length of a five-laps course on a 400 m stadium track on lane 1 as described below. With this calibration, all running and cycling distances were measured as reference values by riding the same course at least twice on the bike (with the tyre pressure kept at its calibration time value given above). Additionally, the true lengths of the running and cycling routes were crosschecked using OpenStreetMap and Google Maps. These three distance results per course, i.e., by bike ride, OpenStreetMap and Google Maps, differed only marginally in the second decimal place (3.41 km vs. 3.4(0) km and 36.84 km vs. 36.8(0) km). Notably, this small difference was inevitable as the bike computer’s display provided two decimal places, whereas the map material yielded only one.

For each run, 4 smartwatches were worn simultaneously to reduce the number of runs: two on the left forearm and two on the right forearm. The positions were numbered—(T1) proximal forearm, left; (T2) distal forearm, left; (T3) distal forearm, right; and (T4) proximal forearm, right—and noted for each watch and trial. Five runners (3 males, 2 females) ran 4000 m on a 400 m standard tartan stadium track. The runners were instructed to stay in lane one, which is exactly 400 m in length, at a distance of 30 cm from the inner line [45]; running was maintained in this lane (with the exception of, at maximum, 6 quick overtaking manoeuvres per 4000 m). Additionally, an outdoor run of 3.41 km on an asphalt road in profiled terrain was performed. The tests on the stadium track and the asphalt road were performed in triplicate by five runners, each with four smartwatches in positions T1–T4, resulting in 5 × 3 × 4/10 = 6 separate measurements per watch. The watches were rotated among the runners, as well as the forearm position, after each run. For the cycling measurements, four cyclists (three males, one female) completed a fixed road bike course of 36.84 km on asphalt roads in both directions with five smartwatches attached to the handlebars of each bike or to the cyclist’s forearms (the latter if a non-zero pulse rate was required for correct operation), resulting in 4 × 5 × 2/10 = 4 separate measurements per smartwatch model.

### 2.5. Tracking of Pool Swimming Activities

Three different swimming protocols were performed with the smartwatches in a 50 m Olympic-size pool to determine the accuracy of swimming lap counts and stroke rates. The following measurement protocols were carried out three times per watch, divided among six swimmers (three males, three female), with a different watch worn on each forearm: (1) 200 m individual medley (butterfly–backstroke–breaststroke–front crawl) with stroke transitions every 50 m, (2) 400 m freestyle (performed as front crawl), (3) 200 m individual medley with stroke transitions every 25 m. During the freestyle sequences, the swimmers were filmed by a moving camera to provide a video-based independent validation of the number of strokes detected by the smartwatches. 

### 2.6. Data Analysis and Statistics

The data collected from all smartwatches was transferred to the mobile phone apps of the respective manufacturers via Bluetooth. The following apps were used: Fitbit (Fitbit, Inc., San Francisco, CA, USA), Fossil Smartwatch (Fossil, Inc., Richardson TX, USA), Garmin Connect (Garmin AG, Schaffhausen, Switzerland), Apple Health (Apple, Inc., Cupertion, CA, USA), Huawei Health (Huawei, Shenzhen, Guangdong, China), Mi Fitness (Xiaomi, Inc., Beijing, China), Polar Flow (Polar Electro, Inc., Kempele, Finnland), Samsung Health (Samsung Electronics Co., Ltd., Seoul, South Korea), and Zepp (Zepp North America, Inc., Irvine, CA, USA). Afterwards, the data were transferred manually to Microsoft^®^ Excel^®^ for Microsoft 365 (Microsoft Corporation, Redmond, WA, USA). All mathematical analyses and statistical tests were performed with Microsoft^®^ Excel^®^ for Microsoft 365 (Microsoft Corporation, Redmond, USA), MATLAB R2023a (MathWorks Inc., Natick, MA, USA), and JASP (JASP Team (2023), Version 0.17.1, Amsterdam University, Amsterdam, Netherlands).

To evaluate the accuracy of the heart rate measurement, the mean heart rate and the peak heart rate were each determined for 3 min intervals and then compared to the heart rate values from the chest strap reference measurements. Average and peak heart rate values for the 3 min intervals were directly reported by the smartwatch under study and the chest strap reference, respectively, so that no further averaging or peak detection had to be conducted. The 3 min interval per stage was chosen because of its commonality and significance in performance diagnostics. Derived descriptive statistical parameters comprised minimum deviation (from reference), maximum deviation, mean absolute error, mean absolute percentage error, median, and interquartile range. Pearson and Spearman correlation coefficients were calculated between measured and reference heart rates, along with their coefficients of determination (*R*^2^) and levels of significance. 

To verify the accuracy of the GNSS-based distance measurements, the distances measured by the smartwatches were compared to the true reference distance. In particular, descriptive statistics included the arithmetic mean, minimum, maximum, mean absolute error, mean absolute percentage error, standard deviation, and interquartile range. To test for statistical significance of possible deviations between measured distances and the reference value, t-tests were carried out with effect sizes characterised by Cohen’s *d*. To compare the accuracy of the watches among each other, a one-way repeated-measures ANOVA was conducted with the smartwatch model being the independent variable. In addition, a t-test was conducted for the 4000 m stadium runs to investigate if wearing the watch on the left or right forearm (i.e., inside or outside the lane) had an impact on the measured distance.

For the swimming tests, the number of metres swum by the participants, the number of strokes used, and the SWOLF index were evaluated. The SWOLF value is the time in seconds plus the number of strokes required to swim a given distance, i.e., SWOLF = time in seconds/lap + strokes/lap. Descriptive statistics comprised the arithmetic mean, minimum, maximum, mean absolute error, mean absolute percentage error, standard deviation, and interquartile range.

For all statistical tests, the level of significance was set to *p* < 0.05. If not stated otherwise, results are given in terms of mean ± standard deviation (SD).

## 3. Results

### 3.1. Heart Rate

The mean absolute errors (MAEs) of the average heart rate measurements, as measured for all of the smartwatches, were between 4.2 and 11.8 beats per minute (bpm). This corresponded to mean absolute percentage errors (MAPEs) of 3.1% to 8.3%. The coefficient of determination of the average heart rate was between 0.220 and 0.705. The Spearman correlation coefficients ranged from 0.436 to 0.841 (*p* < 0.001), indicating medium to large associations. In detail, the model-specific results can be found in Table 3, and the respective box plots and Bland–Altman plots are shown in Figure 2, with the regression analysis in Figure 3. All regression lines have a lower gradient than the reference lines. Reviewing the data for all 30 runners, the MAEs for each runner and all 10 watches combined ranged between 1.1 and 19.1 bpm, which corresponds to MAPEs of 0.8–17.0%.

The peak heart rate measurements showed similar MAE values compared to the average heart rate, i.e., between 4.4 and 10.9 bpm, resulting in MAPEs between 3.1% and 7.3%. The coefficients of determination for the peak heart rates were also in a similar range to those of the average heart rate, i.e., between 0.179 and 0.743. The Spearman correlation coefficients equally showed only small differences, ranging between 0.482 and 0.851 (*p* < 0.001). The results are presented in Table 4, the box plots and Bland–Altman plots in Figure 4, and the regression analysis in Figure 5. All regression lines had a lower gradient than that measured by the heart rate monitor. Based on the analysis of the data for all 30 runners, the MAEs for each runner and all 10 watches combined ranged between 0.9 and 29.1 bpm, which is equivalent to MAPEs of 0.6–24.7%. In summary, the peak heart rate measured by the smartwatches showed greater variation than the average heart rate.

The most accurate smartwatches in terms of both the average and the peak heart rates were the APP and the HUA with MAPEs of 3.1%/3.1% and 3.3%/3.4%, respectively.

### 3.2. GNSS-Based Distance

#### 3.2.1. Stadium Track Running Tests

The results of the 4000 m track running tests are summarised in Table 5 and Figure 6. Altogether, seven smartwatches varied, on average, by less than 80 m (<2%) from the true distance. The mean MAPEs were between 0.8% and 12.1%. The FIT severely underestimated the true distance in every measurement (3515 ± 381.20 m, *p* = 0.026, *t*(5) = −3.12, Cohen’s *d* = −1.272), whereas the FOS (4284 ± 150 m, *p* = 0.006, *t*(5) = 4.66, *d =* 1.903) and XIA (4140 ± 123 m, *p* = 0.038, t(5) = 2.79, *d* = 1.139) returned significantly too high values. For all three watches, the effect sizes in terms of Cohen’s d indicate large mean deviations from the true value. Overall, a strong effect of the smartwatch model on the measured distance was confirmed by the ANOVA (*p* < 0.007, *η*^2^ = 0.509).

The post hoc comparisons showed that 9 (out of 45) deviations from the true distance were significant among the individual smartwatch models (*p* < 0.001–0.013), all of which were for the FIT compared to the other models. Regarding body side dependence, the mean distance measured by all smartwatches when worn on the inner wrist (T1 and T2), was 3946 ± 272 m (*n* = 30), as compared to 4052 ± 249 m (*n* = 30) when worn on the outer wrist (T3 and T4). Comparing these values, the difference of 106 m (2,7%) was not significant (*p* = 0.094).

#### 3.2.2. Hilly Outdoor Running Test on Asphalt

The results of the outdoor running test on a hilly asphalt course with a true length of 3410 m are summarised in Table 6 and Figure 7. In general, these measurements were more accurate than those for the 4000 m track test, with the MAPEs ranging from 0.2% to 7.5%. The most accurate smartwatch, in this respect, was the GAF, which had an MAPE of 0.2%. Three smartwatches had SDs of less than 10 m: APP, GAF, and GAV. Four significant deviations (*p* < 0.001–0.049, *t*(5) = −11.62–4,66,) were found, as well as large effect sizes in terms of Cohen’s *d* of 1.90 to −4.74 for AMA, APP, FOS, and SAM. The ANOVA confirmed that the smartwatch model had a large effect on the measurement accuracy (*p* < 0.001, *η*^2^ = 0.263). The post hoc analysis showed 5 (out of 45) significant differences when comparing the 10 models among each other (*p* < 0.001–0.021), affecting the FIT and POL. Comparing the measurements taken on the inside wrist (i.e., the right wrist in the clockwise direction along the loop) with those on the outside wrist (i.e., the left wrist), the difference of 33 m (1,0%) between 3384 ± 181 m (*n* = 24) and 3417 ± 112 m (*n* = 24) was smaller than that for the 400 m track and not significant either (*p* = 0.195). 

Regarding the tracked elevation profile, all watches that offered this analysis readily via an online platform (i.e., AMA, FIT, GAF, GAV, HUA, and POL) tracked the relative elevation changes adequately throughout the course (Figure 7d). However, AMA and HUA returned strongly imprecise absolute elevations (error of approximately 60 to 120 m).

#### 3.2.3. Road Cycling Course

The results for the road cycling course with a true length of 36.84 km are shown in Table 7. The route and elevation profile are shown in Figure 8. All ten smartwatches underestimated the reference distance cycled by participants but were still more accurate than the GNSS measurements recorded while running, with nine watches exhibiting MAPEs below 1%. The most accurate watch was the POL, with an MAPE of only 0.03%. 

Seven significant deviations were observed for the ten watches (*p* < 0.001–0.023, *t*(2–3) = −44.33–−4.29), with large effect sizes in terms of Cohen’s *d* of −22.167 to −2.144. The three watches that had larger standard deviations, or fewer measurement values that could be analysed (POL, FOS, and FIT) due to self-aborted measurements by the watch were not significant. The ANOVA confirmed that the smartwatch model had a large effect on the measurement accuracy (*p* < 0.001, *η*^2^ = 0.775). The performed post hoc analysis showed 12 (out of 45) significant differences among the 10 models when compared to each other (*p* = <0.001–0.049). Similar to the hilly outdoor running test, absolute elevations and cumulative vertical metres climbed were substantially inaccurate for AMA and HUA (Figure 7e), apparently due to an altitude offset of approximately −84 m and −156 m, respectively, and because of highly noisy or smoothed elevation profiles. 

### 3.3. Pool Swimming

Altogether, the smartwatches offer various swimming modes which are compared in Table A1. The FOS, however, lacks a swimming mode; therefore, only nine watches were included in the analysis of the distances tracked during the swimming tests. An evaluation of the 400 m front crawl trial, as presented in Table A2, shows that seven smartwatches recorded correct measurements. Two watches had MAPEs of 4.2% (Fit and POL), as, in each case, one of the three recorded 450 m instead of 400 m.

The results of the 200 m individual medley trial with stroke transitions every 50 m show that the distance tracked had a higher propensity for error when the swimming stroke changed than during a continuous front crawl stroke. Six smartwatches measured the distances correctly, and the three watches that recorded incorrect distance had the following MAPE values: AMA, 8.3%; FIT, 16.7%; POL, 141.7% (Table A3).

For the 200 m individual medley with stroke transitions every 25 m (in the middle and at the end of the lap in the 50 m Olympic-size pool), none of the watches were able to measure the distance correctly (Table A4). The MAPE was between 41.7% and 91.7%.

Five watches provided a functionality to analyse the SWOLF index, i.e., were able to also detect the number of strokes per lap. These values were collected during the 400 m front crawl trial. The evaluation of the number of strokes at 50 m as part of the SWOLF index is shown in Table 8. The MAPE was between 0.4% and 29.5%.

## 4. Discussion

The aim of this study was to compare the accuracy of 10 different smartwatches when measuring the heart rate during running exercises; the distance during running, cycling, and swimming tests; and the stroke rate during swimming trials. 

The results of the heart rate measurements were divided into two parts: the average heart rate and the peak heart rate measurements (Section 3.1). The average heart rates (Figure 2 and Figure 3 and Table 3) had MAPEs between 3.1% and 8.3% while the MAPEs of the peak heart rates (Figure 4 and Figure 5 and Table 4) ranged from 3.1% to 7.3%. The heart rate readings, therefore, were more accurate for the peak values. A factor that could contribute to differences here is the time range, and hence, the number of data points over which the various parameters were calculated. For the peak heart rate, a single maximum spike value was used, while for the average heart rate, values constantly recorded over a 3 min interval were averaged. Another contributing factor could be that the measurement intervals between the watches and the chest strap reference were a few seconds apart in some cases. All regression lines had a gradient lower than the reference for all watches. High values tended to be underestimated, and low values tended to be overestimated. The accuracy of the heart rate measurements performed at the wrist fell short of those achieved with the chest strap. Similar average heart rate MAPEs were recorded compared to values in a study by Chow et al., in which the MAPE deviations in the heart rate measurements during a treadmill exercise were 2.5–8.3%, although that study analysed the heart rate second by second [47].

Interestingly, some runners exhibited only slight deviations from the reference values, and this was consistent for all tested smartwatches. For other runners, all or nearly all of the 10 smartwatches showed substantial deviations. This can be seen in the MAPE values calculated for each athlete, with values of 0.8–17.0% for the average heart rate and 0.6–24.7% for the peak heart rate. If these values for the average heart rate measurements are divided into three ranges—below, within, and above the MAPE values—for the individual smartwatches, the following distribution emerges: nine runners had MAPEs between 0.8% and 3.1%, sixteen runners had MAPEs between 3.2% and 7.7%, and five runners had MAPEs from 10.1% to 17.0%. This indicates that it is not only the accuracy of the smartwatch that influences the result but also the individuality of the runner wearing it. A possible reason could be due to an interference from the photoplethysmography-based measurement concept, whereby an external movement alters the signal strength of the light reflected by the tissue instead of solely the blood flow. In this case, false external frequencies might be measured, such as the step frequency. Future studies could, thus, investigate whether there are non-physiological associations between the supposed measured heart rate and the step frequency.

Regarding the tracking of running distances on the stadium track (Figure 6 and Table 5, Section 3.2.1), three significant deviations for the 4000 m run on the 400 m track were observed: The most accurate watch (GAF) recorded a maximum deviation of 120 m, whereas the least accurate watch (FIT) displayed a deviation of 1020 m. The GAF has a track running mode, and it showed the least deviation, with an MAPE of 0.8%. Overall, the MAPEs were between 0.8% and 12.1%. The shown results fell within the range of MAPEs that were recorded in a study by Budig et al., with an MAPE of 1.8% for one smartwatch model [13] and MAPEs of 1.4–1.9% for two models [18], as well as those of another study by Gilgen-Ammann, with MAPEs of 0.9–4.1% for eight watch models [15].

In addition, the difference between all measurements taken on the inside curve of the wrist (3946 m) and on the outside (4052 m) was 106 m. Considering the geometry of the 400 m track, the theoretical difference would be 7.04 m per lap for a runner using the second lane and not the first. For the 10 laps comprising 400 m each, this difference would be 70.4 m with a track width of 1.22 m [45]. The distance between a runner’s two wrists measured at rest is certainly smaller; however, in motion, the distance between wrists may result in values that differ by up to one lane span as the upper limit. The measured difference of 106 m between the inside-facing watch and the outside-facing watch is, therefore, greater than the difference between lane 1 and lane 2 as 70.4 m over 10 laps. Smartwatch models are available that can differentiate wrist sides, and this could be advantageous for more accurate measurements of distance at curves. Presently, the various manufacturers do not reveal whether they use this information for a more accurate measurement. Future research is indicated to calculate this effect more precisely, even though its practical importance seems limited within the framework of triathlon training.

For the results of the 3410 m hilly running course on asphalt (Figure 7 and Table 6, Section 3.2.2), which showed four significant differences, the mean MAPEs were between 0.2% and 7.5%. Results in a similar range were found for the MAPEs in the study by Budig et al., with 2.8% for one smartwatch [13] and 0.7–4.8% for two models [18], and Gilgen-Ammann reported MAPEs of 3.5–8.5% based on eight watches [15]. However, the measured values varied greatly among some of the watches. With different routes taken in a variety of environments, the MAPEs could not be properly compared because of disturbance variables (e.g., forests, urban canyons, and mountains), but they were in a similar range. The difference in the results between the watches attached facing the inside of the curve and those attached facing outward was 33 m, which is less than that of the 400 m stadium track. However, the outdoor running track had fewer curves than the stadium track.

During road cycling (Figure 8 and Table 7, Section 3.2.3), MAPEs between 0.03% and 4.2% were calculated. Nine models had an MAPE < 1.0% (the exception was the FIT, with only one result). This corresponds to a deviation of fewer than 10 metres per kilometre. This level of accuracy required that the tyre circumference and air pressure, combined with the weight of the rider, be set very precisely using a classic bicycle odometer to achieve a greater accuracy than some of the GNSS measurements. Nevertheless, seven out of nine measurements showed a significant deviation from the true reference distance. This can be explained by the fact that all of the watches, without exception, measured the reference distance as too short on average and, additionally, in almost all of the individual measurements. This can likely be explained by the fact that GNSS distance measurement consists of many individual points. On a curved surface, this means that the measurement consists of many short straight lines as the lower boundary of the actual pathlength of the curved trajectory. At higher speeds (e.g., when cycling), this measurement inaccuracy increases as the short straight lines become longer. Comparing the MAPE values with those in Budig’s studies (MAPE of 0.5% for one smartwatch model and MAPEs of 0.3% for two model) [13,18], nine of the ten watches in this study were in similar ranges. Although they had similar route lengths of 31.5 and 36.7 km, they differed in their locations and, possibly, in the elevation profiles.

As for the tracking of the distances swum in a pool (Section 3.3, Table A2, Table A3 and Table A4 in Appendix A), our findings show that the true distance of 400 metres was measured correctly by seven out of nine of the watches. The MAPEs were between 0.0% and 4.2%, with only two watches measuring one excess lap (FIT and POL). These MAPEs were lower than those recorded in Lee’s study, in which the MAPEs were between 0.0% and 20.6%, split among the different speeds [48]. Two different watches were used in that study. However, similar results were obtained in Budig’s study, reporting MAPEs of 0.4% and 4.6% for two models in the 500 m breaststroke [18]. In the 200 m medley swim, six out of nine watches recorded this distance without error. This corresponds to MAPEs of 0.0–141.7%. Upon the addition of a stroke transition after 25 m in the centre of the lane, no smartwatch recorded the lap count correctly. The stroke transition in the middle of the lap was usually counted as the start of a new lap. This resulted in MAPEs between 41.7% and 91.7%. In another study by Brunner et al., swimming stroke changes performed in the centre of the lap were described as a mixed style, and greater accuracies were also reported [49]. This could not be confirmed with the models in the present study. When measuring the number of strokes, as a main contributor to the SWOLF index, the MAPEs were between 0.4% and 29.5%. These values per lap were recorded for five watches. The range of values are comparable to those in a study by Lee et al., in which the MAPE values were 6.2–17.6% [48].

## 5. Limitations

There are some limitations to the present study that need to be named. First, the heart rate measurements were carried out under controlled laboratory conditions on a motorised treadmill at a constant speed, with no external disturbance factors or changes in the running speed. The heart rate readings were, thus, as expected, mostly constant over time. Moreover, heart rates were evaluated only in terms of the means and maximum values and not by continuous comparison. However, before further field studies can be carried out, tests under laboratory conditions are required.

Second, regarding the GNSS measurements, the number of runs and bike rides per watch was limited due to the availability of athletes at the same time, bearing in mind that the measurements had to be carried out at the same day and roughly the same daytime to ensure identical GNSS satellite conditions for all bouts. Future research should try to increase the number of parallel measurements by increasing the number of identical smartwatch models and athletes available. Furthermore, it could not be determined for all watches which of the global satellite systems—GLONASS, Galileo, or BDS—were used in addition to GPS to determine the position. Despite that, a conclusion can still be drawn as to whether the measured distances are valid.

Third, this study investigated only the accuracy and, thus, the practicability for the three triathlon disciplines during training in a separate manner. A multisport functionality for brick sessions or triathlon competitions could not be evaluated across the smartwatches as it was available only for a small minority of the models tested.

## 6. Conclusions

The results of this study show that the tested smartwatches differed substantially in their accuracies. In particular, optically measured heart rates can deviate considerably from the true values. Comparison with an ECG or a chest strap may be helpful before relying on such wrist-based measurements. As regards distance tracking in running, a conventional GNSS approach still cannot be recommended for measuring the distance run on a 400 m stadium track. Despite all the technical improvements over the past few years, manually or automatically counting the number of laps appears to still be the more accurate approach. However, a smartwatch with specific track mode functionality (as provided by GAF) can overcome this issue and provide sufficiently accurate results for most practical purposes in triathlon and long-distance running exercise. Depending on the smartwatch model, using GNSS for outdoor measurements of distances run or cycled is sufficiently accurate in the context of long-distance running and triathlon exercise control. Not only when measuring personal bests, but in general, errors can occur, and the results should always be critically scrutinised. As for swimming, most of the tested smartwatches were able to record the distance swum in the front crawl 400 m pool swimming trial with sufficient accuracy and were also able to correctly count stroke transitions when performed at the end of a lap. However, our results indicate that current smartwatches are not suitable for the demands of frequent stroke transitions in swim trials. In addition, the stroke rate was reproduced accurately by only a small subset of the watches (HUA and SAM). In essence, all non-temporal values measured by current sports smartwatches should be critically assessed for validity before being used in exercise control, but once their accuracy is confirmed, they can be a useful tool in training management for triathletes and coaches.

## Figures and Tables

**Figure 1 sensors-24-04675-f001:**
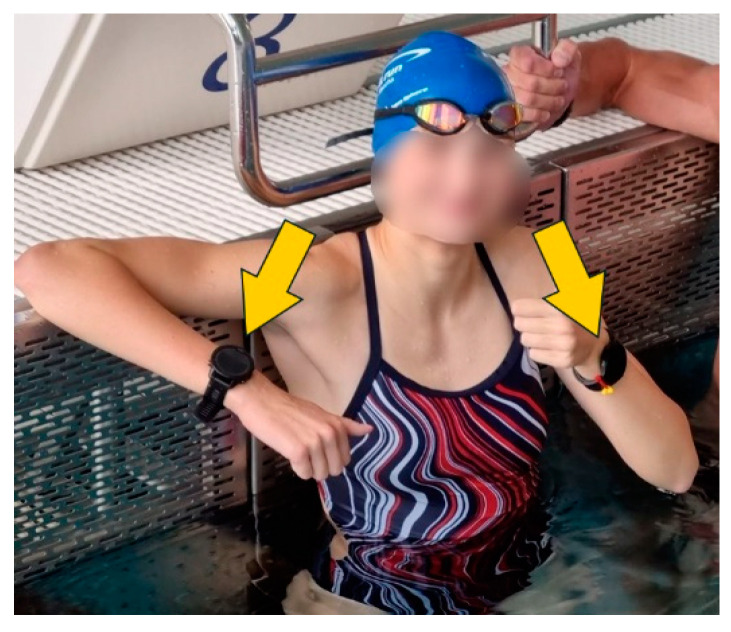
Example of the experimental setup in swimming with two smartwatches tightly worn on both wrists of a female swimmer.

**Figure 2 sensors-24-04675-f002:**
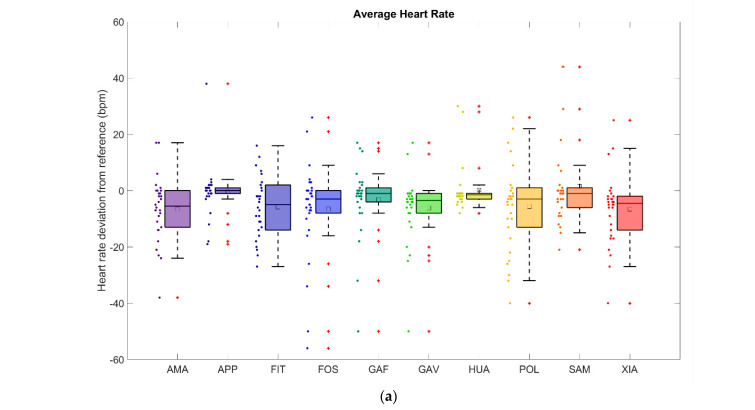
Heart rate measurements: Deviation of the optically measured average heart rates (by photoplethysmography) from the electronic chest strap reference for the ten smartwatches investigated. All values are given in terms of beats per minute (bpm). Throughout this article, each smartwatch model has been assigned a unique colour (e.g. navy for APP, dark cyan for GAF, red for XIA etc.). (**a**) The box plots depict the lowest measured value (bottom dot), the highest measured value (top dot), the median (line inside the box), the first quartile (bottom edge of the box), the third quartile (top edge of the box), and the interquartile range (IQR), where the whiskers are 1.5 times the IQR. Red dots outside the whiskers therefore represent outliers. (**b**) Corresponding Bland–Altman plots.

**Figure 3 sensors-24-04675-f003:**
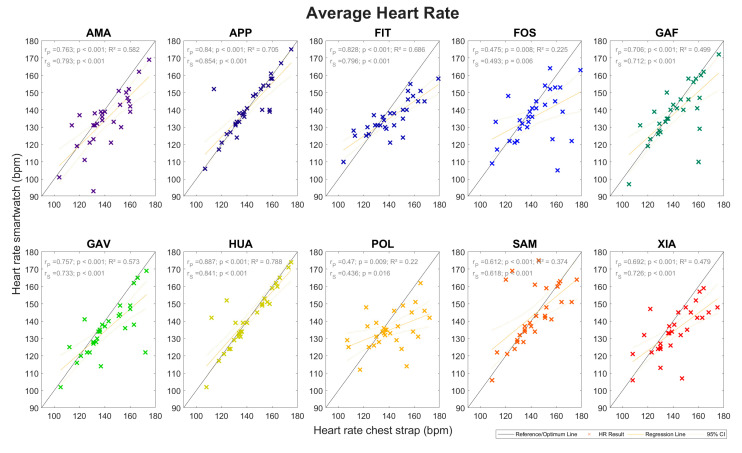
Heart rate measurements: Subjects’ average heart rates as measured by the ten smartwatches (ordinate) versus the chest strap reference (abscissa). The coloured crosses represent each subject’s mean heartrates (chest strap vs. smartwatch), while the red lines show the linear regression results and the grey lines visualise the—always unreached—ideal of perfect agreement. Symbol colours indicate smartwatch models. Symbol meanings include *R*^2^: coefficient of determination; *r*_P_: Pearson’s correlation coefficient; *r*_S_: Spearman’s correlation coefficient; bpm: beats per minute; and CI: confidence interval (95%, red stitched lines).

**Figure 4 sensors-24-04675-f004:**
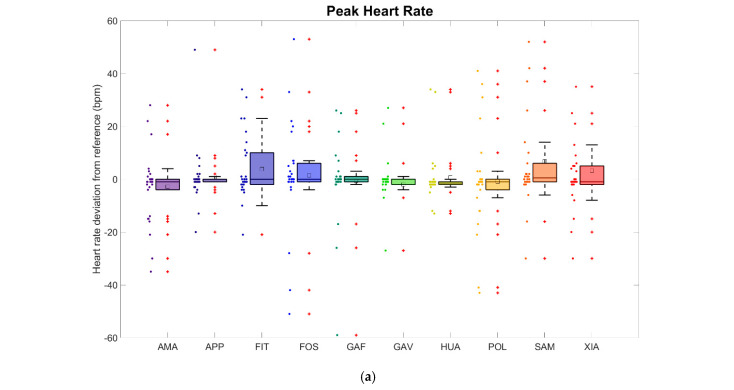
Heart rate measurements: Deviation of the optically measured peak heart rates (by photoplethysmography) from the electronic chest strap reference for the ten smartwatches investigated. Symbol colours indicate smartwatch models. (**a**) Box plots with symbol meanings as in Figure 2. (**b**) Corresponding Bland–Altman plots.

**Figure 5 sensors-24-04675-f005:**
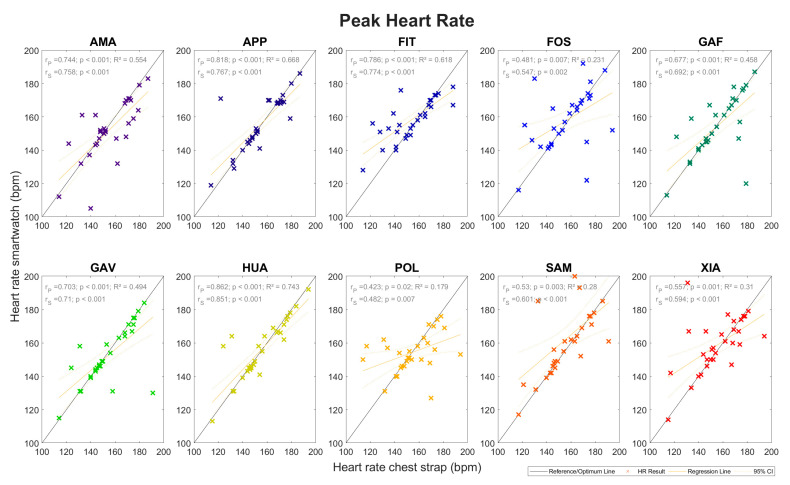
Heart rate measurements: Subjects’ peak heart rates as measured by the ten smartwatches (ordinate) versus the chest strap reference (abscissa). Symbol and colour meanings are the same as those in Figure 3.

**Figure 6 sensors-24-04675-f006:**
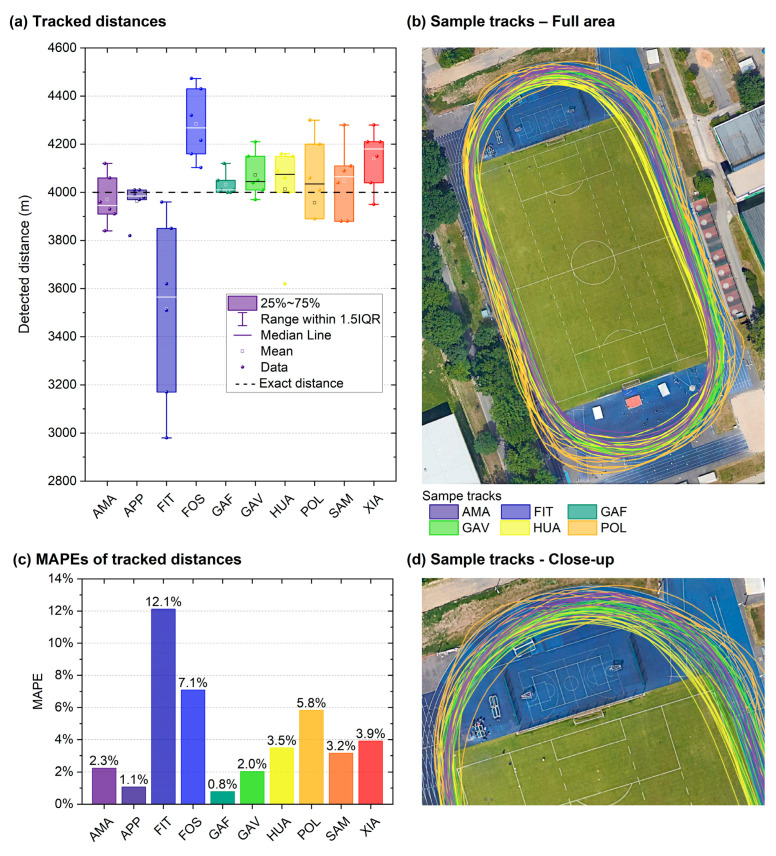
Running tests on a 4000 m stadium track: (**a**) Distances as tracked by ten smartwatches studied, (**b**) a full area overview with sample tracks by six smartwatch models (only a small fraction shown for clarity), (**c**) MAPEs of tracked distances, and (**d**) a close up of the sample tracks. Bar and line colours indicate smartwatch models. Note: For the panels (**b**,**d**), publicly available map data from Google Maps (Google LLC, Mountain View, CA, USA) as of 3 June 2022 were used.

**Figure 7 sensors-24-04675-f007:**
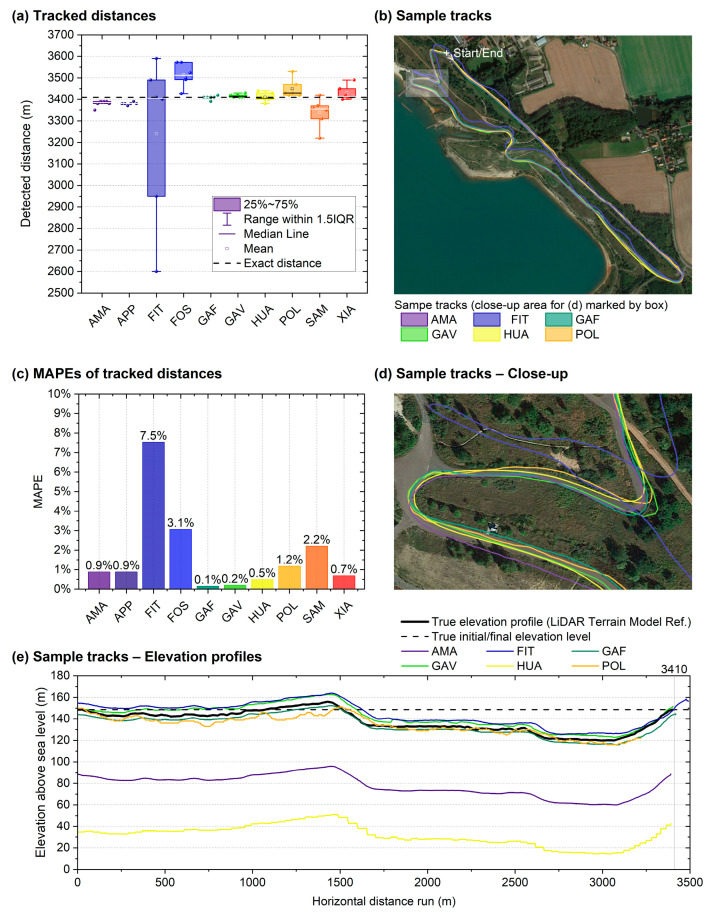
Running tests on a hilly 3.41 km outdoor asphalt course: (**a**) Tracked distances of all ten smartwatches, (**b**) sample tracks in the full area with six different sample tracks, (**c**) MAPEs of tracked distances, (**d**) sample tracks shown in a close-up image. (**e**) Elevation profiles of sample tracks shown in (**b**,**d**) vs. true profile. Bar and line colours indicate smartwatch models. Notes: The reference profile was obtained from publicly available high-resolution LIDAR elevation data for Western Europe based on GeoBasis-DE/BKG using the software tool GPS Visualizer in its current version as of 2019 [46]. For the panels (**b**,**d**), publicly available map data from Google Maps (Google LLC, Mountain View, CA, USA) as of 9 September 2021 were used.

**Figure 8 sensors-24-04675-f008:**
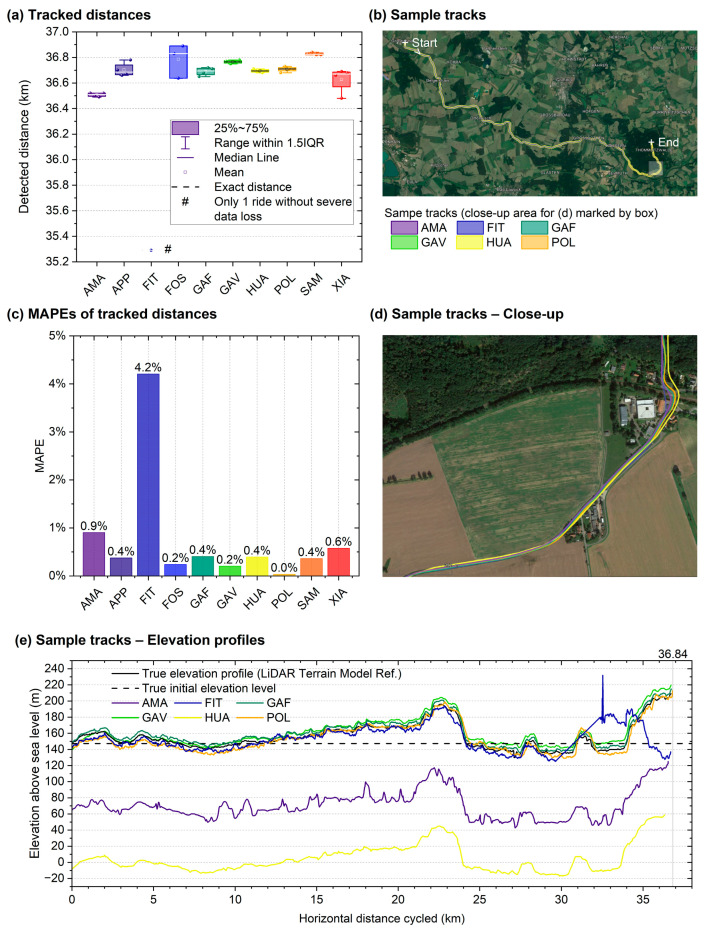
Road cycling course: (**a**) Tracked distances of all ten smartwatches, (**b**) sample tracks in the full area with six different sample tracks (one-way), (**c**) MAPEs of the tracked distances, (**d**) sample tracks shown in a close-up image. (**e**) Elevation profiles of sample tracks shown in (**b**,**d**) vs. true profile. Bar and line colours indicate smartwatch models. Notes: The reference profile was obtained from publicly available high-resolution LIDAR elevation data for Western Europe based on GeoBasis-DE/BKG using the software tool GPS Visualizer [46]. For the panels (**b**,**d,**) publicly available map data from Google Maps (Google LLC, Mountain View, CA, USA) as of 21 April 2023 were used.

**Table 2 sensors-24-04675-t002:** General characteristics of the participants undergoing heart rate measurement before the treadmill exercise (*n* = 30).

Parameter	Mean ± SD	
Age (years)	29.43	±7.35
Sex	15 males/15 females	
Body mass (kg)	70.16	±8.17
Body height (cm)	175.60	±8.77
BMI (kg/m^2^)	22.75	±2.09
Net weekly training time (h)	7.16	±3.75
RR MAP (mmHg)	91.78	±9.07
RR sys (mmHg)	125.33	±15.69
RR dia (mmHg)	74.42	±7.23
HR (12-channel ECG) (1/min)	61.53	±11.34
Selected speed (km/h)	10.62	±2.09

mmHg: millimetre of mercury (≈Torr); RR: Riva Rocci blood pressure; MAP: mean arterial pressure; sys: systolic; dia: diastolic; SD: standard deviation.

**Table 3 sensors-24-04675-t003:** Heart rate measurements: Average heart rates and their deviations from the chest strap reference.

Watch	δMin	δMax	MAE	MAPE	Median	IQR	*R* ^2^	*r* _P_	*p* _P_	*r* _S_	*p* _S_
	bpm	bpm	bpm	%	bpm	bpm					
AMA	0	38	9.37	6.66	−5.5	13.25	0.582	0.763	<0.001	0.793	<0.001
APP	0	38	4.23	3.12	0.0	2.25	0.705	0.840	<0.001	0.854	<0.001
FIT	0	27	9.67	6.68	−5.0	16.00	0.686	0.828	<0.001	0.796	<0.001
FOS	0	56	10.97	7.32	−3.0	8.75	0.225	0.475	0.008	0.493	<0.001
GAF	0	50	7.27	5.02	−1.0	5.75	0.499	0.706	<0.001	0.712	<0.001
GAV	0	50	8.17	5.59	−3.5	7.00	0.573	0.757	<0.001	0.733	<0.001
HUA	1	30	4.30	3.34	−1.5	2.00	0.788	0.887	<0.001	0.841	<0.001
POL	0	40	11.83	8.30	−3.0	15.00	0.220	0.470	0.009	0.436	<0.001
SAM	0	44	8.80	6.35	−1.0	7.75	0.374	0.612	<0.001	0.618	<0.001
XIA	0	40	10.20	7.13	−4.5	12.5	0.479	0.692	<0.001	0.726	<0.001

δMin, δMax, MAE, Median and IQR are in beats per minute (bpm; absolute values). δMin: minimum deviation; δMax: maximum deviation; MAE: mean absolute error; MAPE: mean absolute percentage error; IQR: interquartile range; *R*^2^: coefficient of determination; *r*_P_: Pearson’s correlation coefficient; *p*_P_: type I error probability of Pearson’s correlation coefficient; *r*_S_: Spearman’s correlation coefficient; *p*_S_: type I error probability of Spearman’s correlation coefficient.

**Table 4 sensors-24-04675-t004:** Heart rate measurements: Peak heart rates and their deviations from the chest strap reference.

Watch	δMin	δMax.	MAE	MAPE	Median	IQR	*R* ^2^	*r* _P_	*p* _P_	*r* _S_	*p* _S_
	bpm	bpm	bpm	%	bpm	bpm					
AMA	0	35	8.00	5.31	−1.0	6.75	0.554	0.744	<0.001	0.758	<0.001
APP	0	49	4.37	3.07	0.0	1.25	0.668	0.818	<0.001	0.668	<0.001
FIT	0	34	7.77	5.51	0.0	12.25	0.618	0.756	<0.001	0.774	<0.001
FOS	0	53	10.37	6.81	0.0	7.25	0.231	0.481	0.007	0.547	<0.001
GAF	0	59	6.77	4.34	0.0	2.0	0.458	0.677	<0.001	0.692	<0.001
GAV	0	61	5.67	3.60	0.0	2.0	0.494	0.703	<0.001	0.710	<0.001
HUA	0	34	4.87	3.41	−1.5	1.0	0.743	0.862	<0.001	0.851	<0.001
POL	0	43	10.70	7.28	−1.0	5.5	0.179	0.423	0.02	0.482	<0.001
SAM	0	68	10.93	7.30	0.5	8	0.280	0.530	0.003	0.601	<0.001
XIA	0	65	9.47	6.47	−1.0	7.25	0.310	0.557	<0.001	0.594	<0.001

Abbreviations are as defined in Table 3.

**Table 5 sensors-24-04675-t005:** Running tests on 4000 m track: Deviations of the tracked distances from the true distance for the ten smartwatches investigated.

Watch	Mean	Min.	Max.	SD±	MAE	MAPE	Median	IQR	*p*	*d*
	m	m	m	m	m	%	m	m		
AMA	3970	3840	4120	102.76	90.00	2.25	3945	182.5	0.507	−0.292
APP	3963	3820	4120	72.02	43.33	1.08	3985	77.5	0.268	−0.509
FIT	3515	2980	3960	381.20	485.00	12.13	3565	755	0.026 *	−1.272
FOS	4284	4103	4473	149.06	283.67	7.09	4268	295	0.006 **	1.903
GAF §	4032	4000	4120	47.08	31.67	0.79	4010	67.5	0.160	0.637
GAV	4072	3970	4210	90.42	81.67	2.04	4045	165.0	0.110	0.793
HUA	4013	3620	4160	201.56	140.00	3.50	4075	247.5	0.878	0.066
POL	3957	3280	4300	361.37	233.33	5.83	4035	487.5	0.781	−0.120
SAM	4047	3880	4280	152.27	126.67	3.17	4065	272.5	0.487	0.306
XIA	4140	3950	4280	122.96	156.67	3.92	4180	210.0	0.038 *	1.139

Min: minimum; Max: maximum; MAE: mean absolute error; MAPE: mean absolute percentage error; SD: standard deviation; IQR: interquartile range; *d*: Cohen’s *d*. Levels of significance: * *p* < 0.05, significant; ** *p* < 0.01, very significant; *** *p* < 0.001, extremely significant. § The GAF was used in the track running mode.

**Table 6 sensors-24-04675-t006:** Running tests on hilly outdoor asphalt course: Deviations in the tracked distances from the true distance of 3.41 km for the ten smartwatches investigated.

Watch	Mean	Min	Max	SD±	MAE	MAPE	Median	IQR	*p*	*d*
	m	m	m	m	m	%	m	m		
AMA	3380	3350	3390	15.49	30.00	0.88	3385	17.50	0.005 **	−1.936
APP	3380	3370	3390	6.32	30.00	0.88	3380	5.00	<0.001 ***	−4.743
FIT	3240	2600	3590	382.94	256.67	7.53	3405	652.50	0.326	−0.444
FOS	3515	3427	3572	54.91	104.50	3.06	3512	96.25	0.006 **	1.903
GAF	3408	3390	3420	9.83	5.00	0.15	3410	7.50	0.695	−0.170
GAV	3417	3410	3430	8.16	6.67	0.20	3415	12.50	0.102	0.816
HUA	3410	3380	3440	21.91	16.67	0.49	3405	37.50	1.000	0.000
POL	3450	3420	3530	43.36	40.00	1.17	3430	65.00	0.073	0.923
SAM	3338	3220	3420	67.95	75.00	2.20	3355	95.00	0.049 *	−1.055
XIA	3430	3400	3490	34.06	23.33	0.68	3415	52.50	0.210	0.587

Symbol meanings are the same as in Table 5.

**Table 7 sensors-24-04675-t007:** Road cycling course: Deviations in the tracked distances from the true distance of 36.84 km for the ten smartwatches investigated (*n* = 4).

Watch	Mean	Min.	Max.	SD±	MAE	MAPE	Median	IQR	*p*	*d*
	m	m	m	m	m	%	m	m		
AMA	36,508	36,490	36,520	15.00	332.50	0.90	36,510	27.50	<0.001 ***	−22.167
APP	36,703	36,660	36,780	54.39	137.50	0.37	36,685	97.50	0.015 *	−2.528
FIT §	35,290	35,290	35,290	-	1550	4.21	35,290	-		
FOS §	36,786	36,637	36,890	132.20	87.67	0.24	36,830	253.00	0.550	−0.411
GAF	36,690	36,650	36,720	31.62	150.00	0.41	36,695	60.00	0.002 **	−4.743
GAV	36,765	36,750	36,780	12.91	75.00	0.20	36,765	25.00	0.001 **	−5.809
HUA	36,695	36,680	36,710	12.91	145.00	0.39	36,695	25.00	<0.001 ***	−11.232
POL	36,628	36,480	36,690	99.12	12.50	0.03	36,670	162.50	0.080	−1.306
SAM	36,708	36,680	36,730	20.62	132.50	0.36	36,710	37.50	0.001 **	−6.427
XIA	36,828	36,820	36,840	9.57	212.50	0.58	36,825	17.50	0.023 *	−2.144

Symbol meanings are the same as those in Table 5. § FIT, *n* = 1; FOS, *n* = 3.

**Table 8 sensors-24-04675-t008:** Deviations in the number of strokes calculated for one 50 m lap during a 400 m front crawl swimming test (*n* = 5).

Watch	δMin	δMax	MAE	MAPE	Median	IQR
				%		
GAF	1.0	2.0	1.5	6.32	1.5	1.0
GAV	1.0	3.0	2.1	8.81	2.5	1.5
HUA	0.0	1.5	0.5	2.14	0.5	1.0
POL	0.5	28	8.5	29.45	2.5	18
SAM	0.0	0.5	0.1	0.43	0.0	0.25

δMin, δMax, MAE, Median and IQR are in metres (absolute values). δMin: minimum deviation, δMax: maximum deviation, MAE: mean absolute error, MAPE: mean absolute percentage error, IQR: interquartile range.

## Data Availability

Data are contained within the article.

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
