# Peer review of "Validity of Current Smartwatches for Triathlon Training: How Accurate Are Heart Rate, Distance, and Swimming Readings?"

_sensors, 2024, doi:10.3390/s24144675_

Round 1
Reviewer 1 Report
Comments and Suggestions for Authors
I read the paper with interest. The topic undertaken is attractive for the audience, considering that not only professionals but also amateurs train in triathlon disciplines. The scientific rigor of the methodology as well as results presentation, guarantee the possibility of experiment reproduction.
The following minor suggestions may improve the quality of the manuscript:
1) For clarity, abbrevation instead of full model name should be used in figures (Fig.1, 3, 5, 6, 7).
2) For comparison purposes with other studies, I would recommend changing Fig.2 & 4 from correlation analysis/regression to Bland-Altman. In validation, it is the most popular and acceptable form of results presentation.
3) Unfortunately Fig. 2 & 4 in current form have a low quality (greyscale, small font etc.).
4) Please describe more precisely how you determined mean (from five trials in each subject?) and maximal heart rate. I propose to resign from a maximal value in results presentation because in effort with constant speed, your results are really "spikes”, not maximal. The maximal is expected in increasing workloads.
Author Response
Please see attached PDF file.

Reviewer 2 Report
Comments and Suggestions for Authors
The research is highly worth for people who live with wearable technologies. The research made results will help many people's lives and exercise. This paper is in nearly publishable condition, except for a few minor revisions. The experiment was quite laborious and required effort.
Authors must ensure that it is OK disclosing their brand name legally.
Make all figures are visible if you want to use figures.
Author need to guide data in each discussion. Or, author can distribute the discussion section into each results. Because, there are many data providing worth information. However, it is difficult to match each data with discussion.
Author may provide key hardware specifications of models.
Author needs to provide picture or schematic of user scene. Sometimes, sensors in watch doesn't work well if the watch is not tightly contact to the human skin.
Author Response
Please see attached PDF file.
